# CTSyn: A Foundation Model for Cross Tabular Data Generation

**Xiaofeng Lin[1], Chenheng Xu[1], Matthew Yang[2], Guang Cheng[1]**

[1]Department of Statistics and Data Science, University of California Los Angeles CA, USA
[2]Department of Computer Science, University of California Los Angeles, CA, USA

{bernardo1998,chenhengx0101,ymatt24,guangcheng}@ucla.edu

## Abstract

Generative Foundation Models (GFMs) have achieved remarkable success in producing high-quality synthetic data for images and text. However, their application to tabular data presents significant challenges due to the heterogeneous nature of table features. Current cross-table learning frameworks struggle because they lack a generative model backbone and an effective mechanism to decode heterogeneous feature values. To address these challenges, we propose the Cross-Table Synthesizer (CTSyn), a diffusion-based generative foundation model for tabular data generation. CTSyn comprises two key components. The first is an autoencoder network that consolidates diverse tables into a unified latent space. It dynamically reconstructs table values using a table schema embedding, allowing adaptation to heterogeneous datasets. The second is a conditional latent diffusion model that generates samples from the learned latent space, conditioned on the table schema. Through large-scale pre-training, CTSyn outperforms existing table synthesizers on standard benchmarks in both utility and diversity. These results position CTSyn as a promising framework for synthetic table generation and lay the groundwork for developing large-scale tabular foundation models.

## 1 Introduction

Generative Foundation Models (GFMs) have revolutionized fields such as Computer Vision (CV) and Natural Language Processing (NLP)(Bommasani et al., 2021; He et al., 2016; OpenAI, 2023; Touvron et al., 2023; Ramesh et al., 2022; Rombach et al., 2022). Trained on vast datasets (Merity et al., 2016; Deng et al., 2009; Schuhmann et al., 2022) and with versatile model backbones (Vaswani et al., 2017; Ho et al., 2020), these models excel across a diverse range of domains and tasks. They can generate valuable synthetic training examples to enhance the performance of various downstream applications (Kirillov et al., 2023; Li et al., 2023b; Moor et al., 2023; Trabucco et al., 2023; Zhang et al., 2023a).

GFMs also hold immense potential for generating tabular data, a modality essential to many real-world applications (Dash et al., 2019; Borisov et al., 2022; Shwartz-Ziv & Armon, 2022). Despite the ubiquity of tabular data, obtaining high-quality samples for modeling remains a challenge. Although tabular data synthesizers have increasingly gained attention (Xu et al., 2019; Kotelnikov et al., 2023; McKenna et al., 2022), they yield little performance improvement in downstream models when real data is limited (Elor & Averbuch-Elor, 2022; Manousakas & Aydöre, 2023).This limitation stems from a fundamental constraint: synthesizers cannot generate information beyond what is present in the original training data. Tabular GFMs have the potential to overcome this limitation by leveraging diverse pre-training data.

Despite these opportunities, implementing tabular GFMs remains particularly challenging and largely overlooked due to the heterogeneity of column structures, feature sets, and value ranges (Onishi et al., 2023; Huang et al., 2020; Borisov et al., 2022; Zhu et al., 2023; van Breugel & van der Schaar, 2024). Existing methods for transferable tabular learning either model tables with language models (Ye et al., 2024; Wang & Sun, 2022; Hegselmann et al., 2023; Yan et al., 2024) or attempt to learn a unified latent space across datasets (Wang & Sun, 2022; Onishi et al., 2023; Zhu et al., 2023;

Ye et al., 2023). These methods either lack generative capability or rely on pre-trained language models that process tables as unstructured text, distorting structural information and metadata that are critical for effective tabular modeling.

To address these limitations, we propose *CTSyn*, a foundational model specifically designed for generating heterogeneous tables.

- **Unified table representation and reconstruction:** We developed a cross-tabular autoencoder that embeds heterogeneous table rows, projects them into a unified latent space, and decodes tabular values using table metadata as guidance. This approach enables model training across diverse tabular formats, overcoming data-specific structural constraints while preserving structural information in a table-consistent manner.

- **Generative foundation model:** Our versatile conditional diffusion transformer backbone efficiently samples from the unified latent space, enhancing flexibility and applicability across diverse tabular domains.

- **Cross-tabular pre-training:** We conduct extensive cross-tabular pre-training on a large-scale web-scale dataset containing 5 million rows. With diverse data domains covering common table applications, this pre-training serves as a foundation for various downstream generation tasks.

Through extensive benchmarking with real-world datasets, we demonstrate that CTSyn extends the pre-training/fine-tuning paradigm to tabular data generation, achieving state-of-the-art (SOTA) performance on low-data regimue. Crucially, by effectively leveraging prior knowledge and incorporating table metadata, our model unlocks unprecedented potential in synthetic tabular data generation and holds immense promise for extending to various tabular tasks such as regression and classification.

## 2 RELATED WORK

### 2.1 TRANSFERABLE TABLE REPRESENTATION

Self-supervised learning can significantly improve representation quality for various downstream tasks (Gururangan et al., 2020; Yuan et al., 2021b; Wei et al., 2021; Chen et al., 2024). In the tabular domain, methods like VIME (Yoon et al., 2020) train an encoder using a combination of supervised reconstruction loss and mask-array prediction loss, while SCARF (Bahri et al., 2022) employs contrastive loss by utilizing randomly corrupted feature vectors as positive pairs. Subtab (Ucar et al., 2021) and SSP (Chitlangia et al., 2022) integrate contrastive and reconstruction losses. However, these approaches do not produce transferable representations across tables, as they rely on data-specific feature encodings and structures.

Xtab (Zhu et al., 2023) and TabRet (Onishi et al., 2023) introduce transformer-based backbones with separate data-specific featurizers or projection heads for each downstream task. These models achieve transferability at the expense of increasing model complexity as the number of datasets and tasks grows.

Pre-trained Language Models (PLMs) can be used to unify representation dimensions of heterogeneous features. TransTab (Wang & Sun, 2022) extends Subtab's methodology by tokenizing and then encoding column names and categories, creating a latent space that can be shared across tables. By combining tokenization with a masked-value prediction objective, transformer-based models can be trained to perform predictive tasks across tables (Ye et al., 2024; Yak et al., 2023; Yang et al., 2024; Yan et al., 2024). However, such methods include neither a generative model backbone nor a fixed-dimensional row representation and decoder that can be integrated with other generative models.

Another line of research involving PLMs converts tabular features to sentences and models regression/classification problems as NLP tasks (Dinh et al., 2022; Hegselmann et al., 2023; Borisov et al., 2023; Liu et al., 2022; Zhang et al., 2023c;b). Despite enabling transfer learning, these methods face challenges in accurately modeling continuous values and tend to overlook the intrinsic structural properties of tables (van Breugel & van der Schaar, 2024).

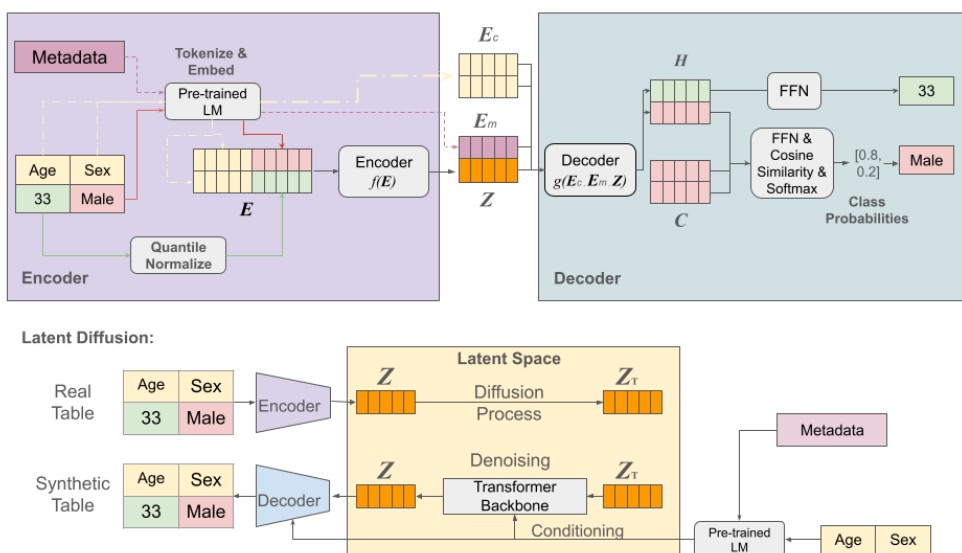

Figure 1: Overview of the proposed CTSyn framework.

## 2.2 SYNTHETIC TABULAR DATA GENERATION

Synthetic Tabular Data Generation (STDG) has long been studied by statisticians (Nowok et al., 2016; Reiter, 2005; Chawla et al., 2002). Recent advances in deep generative models have significantly pushed its boundaries (Che et al., 2017; Kim et al., 2021; Figueira & Vaz, 2022).

In particular, CTGAN and TVAE (Xu et al., 2019) combine conditional generation with Generative Adversarial Networks (GANs) and Variational Autoencoders (VAEs), along with model-specific normalization, to handle highly imbalanced and non-Gaussian columns. CtabGAN+ (Zhao et al., 2021; 2024) proposes a solution for handling mixed-type and long-tailed variable distributions. Autodiff (Suh et al., 2023) and Tabsyn (Zhang et al., 2024) use a combination of latent diffusion and data-specific autoencoder structures. While these are the most similar works to ours, they lack critical transferable encoding and decoding capabilities. TabDDPM (Kotelnikov et al., 2023) achieves the current state-of-the-art in tabular generation by employing separate diffusion processes for numerical and categorical columns.

Some models also synthesize data with Differential Privacy guarantees (Jordon et al., 2018; Zhang et al., 2017; McKenna et al., 2022). Despite their effectiveness in modeling column distributions, none of the above methods is able to effectively boost the training of machine learning models with synthesized data, greatly limiting their usage in data augmentation (Manousakas & Aydöre, 2023).

GReaT (Borisov et al., 2023) and Tabula (Zhao et al., 2023) generate tables using PLMs by treating table rows as natural language text. Despite showing evidence of transferability, they do not consider cross-table pre-training and generation. Additionally, they introduce the risk of producing out-of-bound examples due to unconstrained sampling of output tokens and face the well-known challenge of modeling numeracy in a discrete token space (Wallace et al., 2019).

## 3 METHODOLOGY

In this section, we outline *CTSyn*, our approach to overcoming the challenges of developing a tabular GFM. Figure 1 provides an overview of the proposed framework.

### 3.1 FEATURE EMBEDDING

Let an observation (row) in a mixed-type table be represented as

$$\boldsymbol{x} = (c_1, x_1, c_2, x_2, \ldots, c_p, x_p),$$

where $c_i$ for $i = 1, 2, \ldots, p$ are the feature names, and $x_i$ for $i = 1, 2, \ldots, p$ are the corresponding feature values, which can be either numerical or categorical. The parameter $p$ denotes the number of features in this row. Let $m_{\text{meta}}$ be the text metadata describing the context of the table.

To facilitate knowledge transfer across tables, it is crucial to develop a unified representation that preserves information at the cell, observation, and table levels. Using techniques such as one-hot encoding and min-max normalization results in the loss of structural and contextual information: tables with entirely different contents can end up with identical representations. Thus, we tokenize and embed all levels of information into vectors of consistent dimensions to facilitate unified modeling.

The first step in this process is consolidating all text metadata, column names, and categorical values, ensuring that integer class labels and abbreviations are expanded into their full textual forms to preserve their original meanings. Then, we create embeddings as follows:

$$e_m = \text{LM}(m), \quad e_{c_i} = \text{LM}(c_i), \quad e_{x_i} = \begin{cases} \text{LM}(x_i) & \text{if } x_i \text{ is categorical,} \\ \text{Quantile}(x_i) \cdot \mathbf{1} & \text{if } x_i \text{ is numerical} \end{cases}$$

where LM is a pre-trained text embedding model that is invoked only once for each unique category or column name. Quantile is a quantile transformer (Pedregosa et al., 2011) fitted to the dataset, mapping numerical values to a uniform distribution. $\mathbf{1}$ is a vector of ones with the same dimension as $M_{\text{LM}}$, ensuring that numerical and categorical embeddings maintain consistent dimensionality.

Finally, we interleave all embeddings and flatten them into a sequence:

$$\boldsymbol{E} = [(e_{c_1}, e_{x_1}), (e_{c_2}, e_{x_2}), \ldots, (e_{c_p}, e_{x_p})] \in \mathbb{R}^{p \times 2M_{\text{LM}}}, \tag{1}$$

where $M_{\text{LM}}$ is the dimensionality of the language model embeddings. Each step in the sequence is formed by concatenating the column name embedding $e_{c_i}$ with the corresponding column value embedding $e_{x_i}$. This interleaving creates cell-level representations that encapsulate both column type and value context, eliminating the need to learn the relative positions of column names and values in the sequence. This design aligns with the permutation-invariant property of tabular data.

## 3.2 AUTOENCODER FOR HETEROGENEOUS TABLES

**Encoder:** To facilitate efficient learning of the diffusion model, we use an encoder model $f$ to further compress the input sequence $\boldsymbol{E}$ into a fixed-dimensional latent vector:

$$\boldsymbol{z} = f(\boldsymbol{E}) \in \mathbb{R}^{\ell \times M_{\text{agg}}},$$

where $\ell$ is the latent dimension, and $M_{\text{agg}}$ is the size of each latent vector. The encoder is based on the Perceiver Resampler (Yuan et al., 2021a), consisting of multi-head attention (MHA) blocks and linear layers. The learnable latent parameters serve as queries, while the keys and values consist of the concatenation of the latent queries and the flattened input sequence $\boldsymbol{E}$.

In each layer of the encoder, a cross-attention operation is performed where the latent queries iteratively attend to both the input sequence (in the first layer) and the latent representations (in subsequent layers). Formally, the output of one attention block is given by:

$$\boldsymbol{Z}^{(l+1)} = \text{FFN}\left(\boldsymbol{Z}^{(l)} + \text{MHA}(q = \boldsymbol{Z}^{(l)}, kv = \boldsymbol{Z}^{(l)})\right),$$

where $\boldsymbol{Z}^{(l)}$ is the latent representation at layer $l$, $\text{MHA}(\cdot)$ represents the multi-head attention operation, and FFN is a feedforward network. At the first layer ($l = 0$), the keys and values are the concatenation of the latent queries and the input sequence, i.e., $kv = [\boldsymbol{Z}^{(0)}; \boldsymbol{E}]$.

Note that we do not include positional embeddings in the sequence $\boldsymbol{E}$ to maintain the permutation invariance property of tabular features.

Following the variational autoencoder (VAE) framework, we use two separate encoders, where each output serves as either the mean vector $\mu \in \mathbb{R}^{\ell \times M_{\text{agg}}}$ or the log-variance vector $\log \sigma^2 \in \mathbb{R}^{\ell \times M_{\text{agg}}}$, respectively. The encoder outputs parameterize the latent distribution, and for each input, we sample the latent vector $\boldsymbol{z}$ using the reparameterization trick, given the predicted mean $\mu$ and log-variance $\log \sigma^2$.

**Decoder with Meta Guidance:** To enable cross-tabular training, the decoder must handle varying column orders and combinations of mixed-type columns while ensuring permutation invariance. Since positional embeddings are excluded during encoding, our approach uses a cross-attention-based transformer decoder, where the column order in the decoder is guided by explicit embeddings of the target column names.

We encode the column names using a PLM as:

$$\boldsymbol{E_c} = [e_{c_1}, e_{c_2}, \ldots, e_{c_p}],$$

where each $e_{c_i}$ represents the embedding of column $c_i$. These embeddings serve as queries for the decoder.

The table metadata embedding $e_m$ is concatenated with the latent variables $\boldsymbol{z}$ (derived from the encoder) to form the keys and values. The decoder $g(\cdot)$ operates by cross-attending to $\boldsymbol{E_c}$ and $[e_m, \boldsymbol{z}]$. The order of the output from the decoder is thus determined by the order of columns in $\boldsymbol{E_c}$, and the model learns to dynamically extract cell information from the row latent representation. The output of the decoder is:

$$h = g(\boldsymbol{E}_c, [e_m; f(\boldsymbol{E})]) \in \mathbb{R}^{p \times M_{\text{decoded}}}.$$

**Table reconstruction:** The decoded embeddings are used to reconstruct the table cell values.

For numerical variables, the decoding process transforms the embedding back into scalar values using a linear layer followed by a sigmoid activation function, producing the predicted value:

$$\hat{x}_i^{\text{num}} = \text{Sigmoid}(\text{Linear}(h_i)).$$

For categorical columns, we use a loss based on cosine similarity to accommodate unseen categories in real-world applications. Inspired by (Yak et al., 2023), we compute the cosine similarity by first refining both the reconstructed embeddings and the PLM embeddings of all real categories using a linear layer. Then, we calculate the cosine similarity between these embeddings, apply softmax to the similarities, and use the resulting distribution as the predicted class probabilities:

$$\hat{P}(x_j^{\text{cat}}) = \text{Softmax}\left(\text{CosineSim}\left(\text{Linear}(h_j), \text{Linear}(\boldsymbol{C})\right)\right),$$

where $h_i$ and $h_j$ are the latent representations of the $i$-th numerical cell and $j$-th categorical cell, respectively, and $\boldsymbol{C}$ represents the set of embeddings for all possible categories in the column. These predicted probabilities and values are then used to reconstruct the original table by mapping the latent space to the appropriate categorical or numerical values for each cell.

**Training VAE:** Following the $\beta$-VAE setup (Higgins et al., 2017), the overall objective is a combination of numerical and categorical reconstruction losses and KL-regularization on the latent space:

$$\mathcal{L} = \sum_{i=1}^{p} \mathcal{L}_{\text{num}}(x_i^{\text{num}}, \hat{x}_i^{\text{num}}) + \sum_{j=1}^{q} \mathcal{L}_{\text{cat}}(x_j^{\text{cat}}, \hat{P}(x_j^{\text{cat}})) + \beta \sum_{k=1}^{\ell} D_{\text{KL}}(\mathcal{N}(\mu_k, \sigma_k^2) \| \mathcal{N}(0,1)),$$

where $\mathcal{L}_{\text{num}}$ is the MSE loss for numerical variables, $\mathcal{L}_{\text{cat}}$ is the cross-entropy loss for categorical variables, $D_{\text{KL}}$ is the KL-divergence between the learned latent distribution $\mathcal{N}(\mu_k, \sigma_k^2)$ and the standard Gaussian $\mathcal{N}(0,1)$, and $\beta$ is a weighting factor balancing the reconstruction and KL-divergence terms.

**Implementation:** We use four cross-attention layers for both the encoder and decoder, with $\ell = 16$ latent dimensions and $M_{\text{agg}} = 64$. All VAE models in this paper are trained using the AdamW optimizer with an initial learning rate of 0.0002. The learning rate is multiplied by 0.95 if the validation loss does not improve for 10 consecutive epochs.

We use a $\beta$-VAE setup, starting with $\beta_{\text{max}} = 10^{-2}$, and gradually decrease $\beta$ by multiplying it by 0.7 when the reconstruction loss does not improve for 5 consecutive epochs, until reaching a minimum value of $10^{-5}$.

We construct training batches to contain samples from the same source table, improving training efficiency by reducing inter-domain contrast, as unrealistic domain shifts are not representative of real-world applications. We use GTE-large (Li et al., 2023a) as our text embedding model, as it provides robust representations for metadata and categorical features.

### 3.3 Conditional Diffusion Model for Latent Vector Generation

For cross-tabular generation, a conditional latent diffusion model is preferred because table schemas can vary significantly across domains, with different column types and distributions. An unconditional model would struggle to generalize across these diverse formats, making fine-tuning more difficult for domain-specific tasks. We follow the Denoising Diffusion Probabilistic Model (DDPM) formulation to train a conditional diffusion model, with the objective specified below. The input latent variable $z$ is derived from our VAE.

For the denoising objective, we utilize the $v$-parameterization strategy, which is more effective for latent diffusion than the classic noise prediction strategy. We condition the embedding generation on the embedding sequence $[e_m, E_c]$, which encompasses the schema of the desired table. The model is trained with the following loss function:

$$\mathcal{L}(\theta) = \mathbb{E}_{t,(z_{\text{src}}, z_{\text{trg}}),\epsilon} \left[ \lambda_t \left\| \hat{z}_\theta \left( \sqrt{\alpha_t} z_{\text{trg}} + \sqrt{1 - \alpha_t} \epsilon, t, [e_m, E_c] \right) - z_{\text{trg}} \right\|_2^2 \right],$$

where $z_{\text{trg}}$ is the latent variable from the target sequence, and $\alpha_t$ is the noise schedule. Classifier-free guidance is used to improve sample quality, with conditional and unconditional networks jointly trained, where conditioning is dropped with a probability of 0.1 during training.

Following the specifications in Lovelace et al. (2024), our diffusion model uses a pre-LayerNorm transformer architecture with 12 layers, a hidden dimension of 768, learnable absolute positional encodings, and a GeGLU activation function. The noise level is conditioned via a sinusoidal time embedding, which is processed by an MLP and added to the input sequence. Adaptive layer normalization is applied to each feedforward layer, conditioned on the time embedding. We use the AdamW optimizer with a learning rate of 0.0001, a cosine annealing scheduler, a batch size of 256, and 250 sampling steps.

## 4 Experiment

### 4.1 Evaluation Setup

In this section, we evaluate the performance of CTSyn in representing tables and generating informative and diverse synthetic tabular data. Our primary research questions are: **1. Does pre-training on large, general datasets improve the quality of synthetic data generation for downstream tasks? 2. How does the inclusion of metadata and table schema help CTSyn create effective and transferable table representations?**

**Baselines:** We compare our method against a wide array of baselines in synthetic data generation. These include modified SMOTE (Chawla et al., 2002), CTGAN and TVAE (Xu et al., 2019), TabDDPM (Kotelnikov et al., 2023), TabSyn (Zhang et al., 2024), AIM (McKenna et al., 2022), PATE-CTGAN (Jordon et al., 2018), and GReaT (Borisov et al., 2023). The implementation of baselines is detailed in Section B.

**Dataset Construction:** We use a filtered version of the OpenTab dataset (Ye et al., 2024) as our pre-training set. The filtering follows the strategy outlined in (Yan et al., 2024), which excludes duplicate tables, tables containing free-text, date-time, or personally identifiable information (PII) columns, tables with fewer than 10,000 rows, and tables with categorical columns in integer label format that cannot be mapped back to their original string representations. After filtering, the pre-training set comprises 86 tables with a total of 5.01 million observations. Note that while the multi-modal representation framework of CTSyn is extensible to include text and date-time variables, we focus on numerical and categorical variables in this work (an extension to text and date-time is demonstrated in Appendix F).

For downstream benchmarking, we evaluate eleven real-world datasets widely used in the tabular synthesis literature (Suh et al., 2023; Kotelnikov et al., 2023; Zhang et al., 2024) (see Table 1). To avoid any data leakage, we manually verify that none of the datasets used for pre-training appear in the downstream benchmarks; further details on these datasets can be found in Appendix C. For each downstream dataset, we randomly split the data into a fine-tuning set (80%) and a held-out test set (20%). The fine-tuning set is then randomly shuffled, and few-shot subsets are created by

selecting the first 30, 50, 100, 200, and 500 rows, respectively. This procedure allows us to assess the performance of synthetic data generation in low-data scenarios. For each few-shot subset, a separate generator is trained, and three synthetic tables are generated over three independent trials. All synthetic tables generated for a given downstream dataset are evaluated using the same held-out test set. Synthetic data modeled from different subsets are all sampled to 500 observations to ensure a fair comparison and highlight impact of training data size.

For CTSyn, we pre-train the autoencoder for 300 epochs and the diffusion model for 200,000 steps. For fine-tuning, we train the conditional diffusion model and decoder network of the autoencoder while freezing the encoder to maintain alignment in the latent space. We fine-tune the decoder for 100 epochs and the diffusion model for 10,000 steps.

| Dataset | Rows | Target | Num Cols | Cate Cols |
|---|---|---|---|---|
| Faults | 1941 | Classification | 34 | 0 |
| Wilt | 4839 | Classification | 5 | 1 |
| HTRU2 | 17898 | Classification | 8 | 1 |
| News | 39644 | Regression | 60 | 0 |
| Bean | 13611 | Classification | 16 | 1 |
| Obesity | 2111 | Classification | 8 | 9 |
| Titanic | 714 | Classification | 6 | 2 |
| Insurance | 1338 | Regression | 4 | 3 |
| Abalone | 4177 | Regression | 8 | 1 |
| Shoppers | 12330 | Classification | 16 | 2 |
| Indian Liver Patient | 579 | Classification | 9 | 2 |

Table 1: Summary Statistics of Downstream Datasets

**Metadata:** The text data describing table context was generated by querying the ChatGPT-4o (OpenAI, 2024) model with the first 10 rows of each table and prompting it to generate a description of the table's context, domain, and potential use cases. The prompt and metadata generation process is detailed in Appendix G.

## 4.2 STATISTICAL FIDELITY

| Model | Shape | Corr | Precision | Recall |
|---|---|---|---|---|
| SMOTE | 0.96 (0.02) | 0.91 (0.03) | 0.68 (0.04) | 0.02 (0.003) |
| CTGAN | 0.81 (0.04) | 0.73 (0.02) | 0.57 (0.03) | 0.014 (0.002) |
| TVAE | 0.88 (0.03) | 0.89 (0.04) | 0.35 (0.02) | 0.01 (0.001) |
| AIM | 0.63 (0.05) | 0.70 (0.02) | 0.01 (0.001) | 0.03 (0.003) |
| PATECTGAN | 0.15 (0.01) | 0.47 (0.03) | 0.01 (0.001) | 0.02 (0.002) |
| TabDDPM | 0.93 (0.03) | 0.93 (0.02) | 0.59 (0.04) | 0.027 (0.004) |
| TabSyn | **0.97** (0.01) | 0.93 (0.02) | **0.69** (0.03) | 0.003 (0.001) |
| GReaT | 0.90 (0.03) | 0.64 (0.04) | 0.68 (0.03) | 0.005 (0.001) |
| CTSyn | 0.94 (0.02) | **0.95** (0.02) | 0.64 (0.04) | **0.075** (0.006) |

Table 2: Statistical Fidelity Metrics. Scores are averaged across benchmark datasets.

We evaluate the similarity between real and synthetic tables based on the marginal distributions of columns, column-wise correlations, and sample-level coverage.

For column distributions, we use the Kolmogorov-Smirnov (KS) test for numerical columns and Total Variation Distance (TVD) for categorical columns, subtracting them from one so that higher values indicate better similarity. For column-wise correlation, we apply Pearson's correlation for numerical columns, a contingency similarity metric for categorical columns, and a combined method for mixed types (Dat, 2023). For sample-level coverage, we measure precision and recall to quantify the overlap between real and synthetic data (Alaa et al., 2022).

Table 2 presents the average similarity of column distributions and correlations across benchmark datasets. CTSyn consistently matches or exceeds state-of-the-art baselines. While methods like TabSyn and SMOTE excel in maintaining lower-order statistical similarity due to their focus on replicating training data distributions, CTSyn demonstrates superior performance in capturing complex relationships between columns. This is particularly evident in its higher correlation scores and significantly better recall, indicating its ability to preserve important structural relationships within the data, as well as benefiting from the regularization effect of pre-training.

## 4.3 MACHINE LEARNING UTILITY

To evaluating machine learning utility of synthetic data, we fit the following classifiers: logistic regression, Naïve Bayes, decision tree, random forest, XGBoost (Chen & Guestrin, 2016), and Cat-

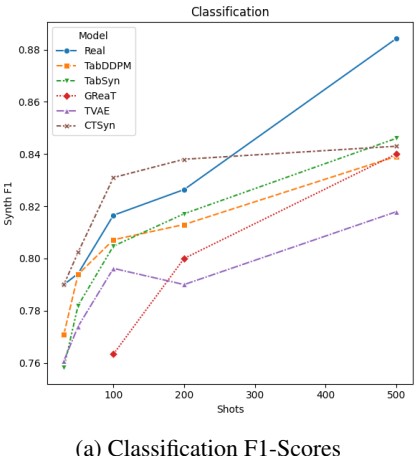

(a) Classification F1-Scores

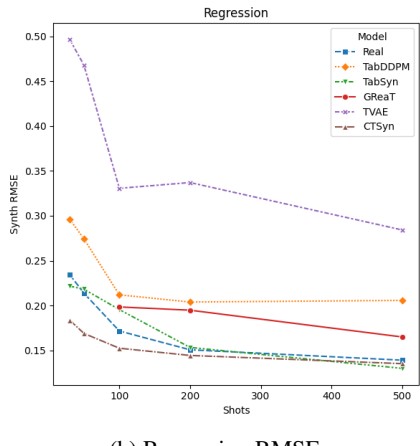

(b) Regression RMSE

Figure 2: Downstream Machine Learning Utility on Classification and Regrssion Datasets, on synthetic data from different generators.

Boost (Prokhorenkova et al., 2018), and then evaluate them on the holdout test set. Figure 2 reports the average classification F1-score (for classification datasets) and regression root mean squared error (RMSE) across models and shots. For clarity, only the top five performing models are shown. We report scores for the remaining models in Appendix E.

We observe that for the low-data regime (Seedat et al.) with $N \leq 100$, CTSyn consistently outperforms all baselines and even real data at the corresponding scale. The performance gap widens in the 100–200 shot range. This indicates CTSyn's ability to leverage pre-training data to assist training when real data is limited. Note that GReaT failed to generate text that follows the tabular format for $N < 100$. However, as the data size increases further, the advantage of CTSyn diminishes. We conjecture that this phenomenon is due to an ineffective transfer learning setup and leave the exploration of transfer learning for tabular GFMs beyond the simple pre-training/fine-tuning paradigm to future work.

| Model | PCT | Authenticity |
|---|---|---|
| SMOTE | 0.82 (0.24) | 0.88 (0.03) |
| CTGAN | 0.84 (0.04) | 0.91 (0.29) |
| TVAE | 0.84 (0.01) | 0.95 (0.31) |
| AIM | 1.00 (0.00) | 1.00 (0.00) |
| PATECTGAN | 1.00 (0.00) | 1.00 (0.06) |
| TabDDPM | 0.85 (0.07) | 0.87 (0.16) |
| TabSyn | 0.80 (0.03) | 0.93 (0.03) |
| GReaT | 0.74 (0.21) | 0.79 (0.03) |
| CTSyn | **0.90 (0.02)** | **0.97 (0.06)** |

Table 3: Privacy scores of synthesized data. Best scores of non-DP synthesizers are bolded.

## 4.4 DIVERSITY AND PRIVACY

We evaluate the diversity and privacy of the synthesized data using two metrics: the proportion of synthetic examples with L2 distance closer to the test set (PCT) compared to the training set (Platzer & Reutterer, 2021), and authenticity scores (Alaa et al., 2022), which assess the likelihood that a synthetic data point is genuinely generated rather than a memorization of real data. Lower PCT or authenticity values suggest that synthetic data points are too close to the training set, raising concerns about potential memorization risk. A powerful generator can easily memorize training data, achieving falsely high fidelity and utility without truly generating new samples, thus harming downstream model generalization and breaching individual privacy.

Table 3 presents the diversity scores. CTSyn achieves the highest PCT and authenticity scores compared to state-of-the-art models like TabDDPM and TabSyn, indicating that CTSyn produces more distinct synthetic data. CTSyn's high PCT scores are comparable to those of AIM and PATE-CTGAN, which incorporate Differential Privacy (DP) mechanisms to reduce proximity to real data. However, these DP models have demonstrated poor fidelity and utility in previous sections. CTSyn,

by contrast, achieves a balance between data utility, diversity, and privacy, further supporting the claim that pre-training acts as regularization.

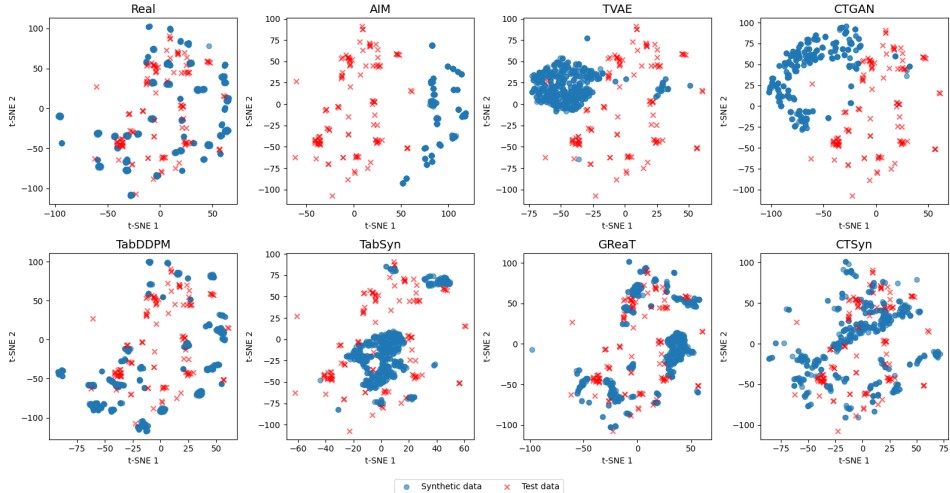

Figure 3: T-sne plot of Indian Liver Patient dataset from different synthesizers.

We further illustrate the diversity of synthesized data using a 2D t-SNE projection in Figure 3 for the Indian Liver Patient dataset. Among all synthesizers, CTSyn generates the most diverse data distribution, covering a wider region around the test set. In contrast, baseline models tend to overfit to regions surrounding certain data points. This diversity, enabled by pre-training, explains CTSyn's superior utility, as the diverse pre-training data serves as implicit regularization, promoting better generalization.

## 4.5 ABLATION STUDY

| Variants | In-distribution | | Unseen Columns | | Permuted Columns | |
|---|---|---|---|---|---|---|
| | MSE | Acc | MSE | Acc | MSE | Acc |
| Column & Meta | 0.0004 | 0.94 | 0.0063 | 0.76 | 0.0005 | 0.94 |
| Column Name Only | 0.0007 | 0.93 | 0.0068 | 0.68 | 0.0006 | 0.93 |
| Meta Only | 0.0008 | 0.91 | 0.0082 | 0.61 | 0.0009 | 0.91 |
| PE Only | 0.006 | 0.87 | 0.07 | 0.54 | 0.08 | 0.67 |

Table 4: Reconstruction performance of different autoencoder settings.

**Importance of Metadata.** We evaluate the key factors for effective cross-tabular representation and reconstruction by testing different autoencoder variations: removing column name embeddings in decoder guidance, removing both column names and metadata embeddings while using positional encoding to encoder on positions of seuqnce $E$. These models are trained on the same pre-training data and tested on three scenarios: (1) in-distribution data from the validation splits, (2) unseen data from downstream test datasets, and (3) Permuted data, where column order is randomly permuted.

Table 4 presents the results in terms of Mean Squared Error (MSE) and categorical accuracy. Removing column names primarily affects categorical column reconstruction, emphasizing the need for contextual representation. The replacement of metadata with PE significantly worsens performance, particularly when handling permuted column order. This highlights a fundamental distinction between tables and unstructured data like text: tables have permutation-invariant structures, and relying on positional information, as in PE, is ineffective.

We further observe that while retaining column names alone results in some performance degradation compared to the full model, the impact is less severe than dropping column names entirely. This outcome highlights the importance of column names in capturing table structure and semantics. We

conjecture that this robustness arises from our diverse pre-training dataset, where column names act as proxies for metadata.

| Model | Pretrained | Shape | Corr | Synth F1 | Synth RMSE | PCT |
|-------|-----------|-------|------|----------|-----------|-----|
| CTSyn | ✓ | 0.94 | 0.95 | 0.84 | 0.13 | 0.97 |
|       | ✗ | 0.96 | 0.90 | 0.80 | 0.17 | 0.90 |
| GReaT | ✓ | 0.90 | 0.71 | 0.79 | 3.75 | 0.88 |
|       | ✗ | 0.90 | 0.64 | 0.83 | 0.18 | 0.79 |
| TabSyn | ✓ | 0.88 | 0.82 | 0.75 | 0.23 | 0.90 |
|       | ✗ | 0.97 | 0.93 | 0.84 | 0.14 | 0.93 |

Table 5: Impact of transfer learning on different models.

**Impact of Pre-training.** We compare different models' potential to leverage pre-training data. We repeat the experiments with CTSyn trained without pre-training, while GReaT and TabSyn are also pre-trained before training on downstream benchmarks. For GReaT, we pool all pre-training data into one dataloader to train a pre-trained distilled GPT-2 and use the resulting model as initialization for downstream benchmark training. For TabSyn, since its model structure is data-specific, we first train separate VAE networks for *each* pre-training table, pool, and zero-pad all embeddings to the same length, and use them to train a latent diffusion model. During downstream training, the VAE embeddings are padded to the same dimension as in pre-training.

As shown in Table 5, the performance of CTSyn degrades without pre-training, though it remains on par with other state-of-the-art models across all dimensions. On the other hand, the change in performance of GReaT and TabSyn with pre-training is mostly negative, indicating their inability to effectively translate knowledge across tabular domains and further reinforcing the need for a model that comprehensively encodes tabular structure information like CTSyn.

## 5 CONCLUSION

In this paper, we introduced CTSyn, a pioneering framework within the realm of Generative Foundation Models (GFMs) for tabular data. Through extensive experimentation on real-world datasets, we demonstrated that CTSyn effectively represents heterogeneous tabular data while leveraging knowledge from diverse pre-trained tables to enhance synthetic data in low-data regimes.

To the best of our knowledge, CTSyn is the first method to integrating large-scale pre-training in tabular data generation via lateng diffusion. Our results highlight its ability to capture complex relationships between table columns, maintain permutation invariance, and generate synthetic data with high fidelity and diversity, surpassing existing state-of-the-art approaches. By incorporating metadata and schema-based conditioning, CTSyn bridges a critical gap in cross-table generalization, allowing for more robust and transferable table representations.

Despite these advances, challenges remain in fully optimizing tabular generative models for broader applications. Future work should explore the potential of conditional diffusion mechanisms and generative pre-training to further enhance tabular tasks such as predictive modeling, imputation, and multi-table learning. Additionally, investigating transfer learning paradigms beyond simple pre-training/full fine-tuning could improve adaptability across diverse tabular domains, unlocking new frontiers in synthetic data generation such as in-context generation and parameter efficient fine tuning for tabular diffusion model.

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

# Appendix

## A  BROADER IMPACT AND LIMITATIONS

A foundational table generator like CTSyn can significantly enhance various application domains, especially where real data is scarce, sensitive, or expensive to obtain, by providing high-quality synthetic tabular data. In healthcare, for example, CTSyn can generate synthetic patient records that maintain statistical fidelity to real data, enhancing the robustness and generalization ability of machine learning models by augmenting datasets with synthetic data.

CTSyn also facilitates data collaboration between parties, such as advertising companies and social media websites. Through conditional generation, CTSyn can augment one party's dataset with essential columns for business analysis without violating privacy laws that prohibit linking individual data points across parties.

However, CTSyn's performance relies heavily on clean, large-scale tabular datasets. The quality of generated data depends on the training data, and any biases or errors can be propagated. This risk can be mitigated by carefully curating high-quality datasets for different domains. Additionally, despite pre-training reducing memorization of downstream data, individuals included in the pre-training data still face privacy risks, complicating the safe gathering of large datasets. This can be mitigated by properly anonymizing or adding noise to public pre-training datasets to ensure privacy before they are used for pre-training.

The requirement of semantically meaningful category names also present challenges for acquiring large-scale training data, as normalized values must be carefully sanitized and converted back to raw form.

## B  BASELINES IMPLEMENTATION

**CTGAN:** We use the official implementation at https://github.com/sdv-dev/CTGAN. We use embedding dimension =128, generator dimension=(256,256), discriminator dimension =(256,256), generator learning rate=0.0002, generator decay =0.000001, discriminator learning rate =0.0002, discriminator decay =0.000001, batch size=500, training epoch = 300, discriminator steps=1, pac size = 5.

**TVAE**: We used the official implementation at: https://docs.sdv.dev/sdv. We used default parameters: class dimensions =(256, 256, 256, 256), random dimensions=100, 64 channels, l2scale=1e-5, batch size=500, training epoch = 300.

**TabDDPM:** We used the official implementation at https://github.com/yandex-research/tab-ddpm. We used 2500 diffusion steps, 10000 training epochs, learning rate = 0.001, weight decay = 1e-05, batch size = 1024.

**AIM:** We use the code implementation at https://github.com/ryan112358/private-pgm, with default parameters: epsilon=3,delta=1e-9,max model size=80

**PATE-CTGAN:** We adapted the implementation posted at: https://github.com/opendp/smartnoise-sdk/blob/main/synth/snsynth, which combines the PATE (Jordon et al., 2018) learning framework with CTGAN. We use epsilon = 3, 5 iterations for student and teacher network, and the same value for other parameters which are shared with CTGAN.

**GReaT:** We used the official implementation at `https://github.com/kathrinse/be_great/tree/main`. We used a batch size of 32. During pre-training, we began with a pre-trained distilgpt2 model and training for 2 millions steps on the combination of pre-training data. We train 200 epochs for each dataset during finetuning.

**TabSyn:** We use the official implementation at `https://github.com/amazon-science/tabsyn`, with default parameters. For pre-training with heterogeneous VAE embeddings, we train its VAE model for each pre-training dataset, zero-pad all embeddings to the same dimension, and then pre-train a diffusion model on such padded embeddings. During downstream training, the VAE embedding of the downstream datasets are padded to the same dimension as in the pre-training. The pre-trained TabSyn is loaded and diffusion training proceed with it as initialization.

**SMOTE:** The original SMOTE algorithm are designed to upsample minority classes. We extend it to perform interpolation for all classes. For each generation, we first randomly select one target class using empirical class frequency as probability. Then we randomly sample one example from the selected class, and generated interpolated examples using number of nearest neighbour $k = 5$. The interpolation weight $\alpha = 0.5$.

## C  BENCHMARK DATASETS

We provide the URL for the sources of each downstream benchmark set considered in the paper.

1. **abalone** (OpenML) : https://www.openml.org/search?type=data&sort=runs&id=183&status=active (Multi class)
2. **Bean** (UCI) : https://archive.ics.uci.edu/dataset/602/dry+bean+dataset (Multi class)
3. **faults** (UCI) : https://archive.ics.uci.edu/dataset/198/steel+plates+faults (Multi class)
4. **HTRU** (UCI) : https://archive.ics.uci.edu/dataset/372/htru2 (Binary class)
5. **indian liver patient** (Kaggle) : https://www.kaggle.com/datasets/uciml/indian-liver-patient-records?resource=download (Binary class)
6. **insurance** (Kaggle) : https://www.kaggle.com/datasets/mirichoi0218/insurance (Regression)
7. **News** (UCI) : https://archive.ics.uci.edu/dataset/332/online+news+popularity (Regression)
8. **Obesity** (Kaggle) : https://www.kaggle.com/datasets/tathagatbanerjee/obesity-dataset-uci-ml (Multi class)
9. **Shoppers** (Kaggle) : https://www.kaggle.com/datasets/henrysue/online-shoppers-intention (Binary class)
10. **Titanic** (Kaggle) : https://www.kaggle.com/c/titanic/data (Multi class)
11. **wilt** (OpenML) : https://www.openml.org/search?type=data&sort=runs&id=40983&status=active (Binary class)

## D  PRETRAINING DATASETS

We show the OpenTab files included in our pre-training, as well as their summary statistics. The classification type dataset are shown in table 6, and regression datasets in table 7.

| File Name | N | Categorical Cols | Numerical Cols |
|---|---|---|---|
| 2736_Shipping | 10999 | 4 | 6 |
| 1366_bankmarketing | 41188 | 11 | 10 |
| 0944_SensorDataResource | 100000 | 1 | 25 |
| 0144_BNG(bridges_version1) | 100000 | 9 | 4 |
| 0062_BNG(page-blocks,nominal,295245) | 100000 | 10 | 1 |
| 0673_BNG(baseball) | 100000 | 1 | 16 |
| 1046_jungle_chess_2pcs_raw_endgame_complete | 44819 | 1 | 6 |
| 0677_COMET_MC_SAMPLE | 89640 | 0 | 5 |
| 1681_Air-Traffic-Data | 15007 | 12 | 4 |
| 1969_CPS1988 | 28155 | 4 | 3 |
| pulsar_data_train | 12528 | 0 | 9 |
| 0666_BNG(primary-tumor) | 100000 | 18 | 0 |
| 0050_BNG(breast-cancer,nominal,1000000) | 100000 | 9 | 1 |
| 0080_BNG(vote) | 100000 | 17 | 0 |
| 1375_MAGIC-Gamma-Telescope-Dataset | 19020 | 1 | 10 |
| 0063_BNG(credit-g,nominal,1000000) | 100000 | 21 | 0 |
| 1431_Beijing-Multi-Site-Air-Quality | 100000 | 2 | 16 |
| bodyPerformance | 13393 | 2 | 10 |
| term_deposit_subscribed33 | 31647 | 9 | 8 |
| 1465_credit | 16714 | 0 | 11 |
| 0077_BNG(heart-statlog,nominal,1000000) | 100000 | 14 | 0 |
| 0761_BNG(autos,1000,10) | 100000 | 10 | 16 |
| 0142_BNG(breast-w) | 39366 | 1 | 9 |
| 0105_kropt | 28056 | 4 | 3 |
| campaign33 | 12870 | 10 | 6 |
| 2149_electricity | 38474 | 1 | 8 |
| 2701_BitcoinHeist_Ransomware | 24780 | 0 | 8 |
| 0772_BNG(lymph,5000,5) | 100000 | 16 | 3 |
| 0059_BNG(colic,nominal,1000000) | 100000 | 23 | 0 |
| 2750_letter-challenge-unlabeled.arff | 10000 | 1 | 16 |
| 0639_jm1 | 10885 | 1 | 21 |
| fusion_experiment | 100000 | 2 | 17 |
| 1690_Malware-Analysis-Datasets-PE-Section-Headers | 43293 | 0 | 5 |
| 0137_BNG(labor) | 100000 | 9 | 8 |
| 1020_Run_or_walk_information | 88588 | 0 | 7 |
| 0070_BNG(glass,nominal,137781) | 100000 | 10 | 0 |
| classifying_document_types_to_enhance_search_and_recommendations_in_digital_libraries_dataset | 11539 | 2 | 5 |
| 0747_BNG(letter,5000,1) | 100000 | 1 | 16 |
| Warehouse_block | 10999 | 4 | 7 |
| 1579_MagicTelescope | 13376 | 1 | 10 |
| 0160_BNG(hepatitis) | 100000 | 14 | 6 |
| 1981_Higgs | 100000 | 0 | 25 |
| 1674_adult | 48842 | 9 | 6 |
| 2687_Diabetes130US | 71090 | 0 | 8 |
| 0057_BNG(mushroom) | 100000 | 23 | 0 |
| 0074_BNG(tic-tac-toe) | 39366 | 10 | 0 |
| 0078_BNG(vehicle,nominal,1000000) | 100000 | 19 | 0 |
| univ.ai_Test Data | 28000 | 6 | 5 |
| flight_delays_train | 100000 | 7 | 2 |
| bank | 11162 | 10 | 7 |
| Firewall_Rule_Classification | 100000 | 1 | 11 |
| Crop_Agriculture_Data_2 | 88858 | 5 | 4 |
| 0711_Stagger1 | 100000 | 4 | 0 |
| 0674_BNG(wine) | 100000 | 0 | 14 |
| 1942_mushroom | 12960 | 9 | 0 |
| 0968_BNG(segment) | 100000 | 20 | 0 |
| bank_customer_survey | 45211 | 9 | 8 |

Table 6: Classification Task Files

| File Name | N | Categorical Cols | Numerical Cols |
|---|---|---|---|
| 1860_Worldwide-Crop-Production | 21165 | 3 | 2 |
| 2711_medical_charges | 100000 | 0 | 4 |
| 0690_BNG(breastTumor) | 100000 | 6 | 4 |
| 2743_Tallo | 100000 | 9 | 12 |
| MAMe_dataset | 37407 | 4 | 4 |
| 2664_diamonds | 53940 | 3 | 7 |
| 2134_Brazilian_houses | 10692 | 0 | 9 |
| 0693_BNG(wine_quality) | 100000 | 0 | 12 |
| 1587_elevators | 16599 | 0 | 17 |
| 2677_fifa | 19178 | 1 | 28 |
| 0940_seattlecrime6 | 52358 | 5 | 3 |
| 1697_AMD-Stock-Prices-Historical-Data | 10361 | 0 | 6 |
| 1905_New-Delhi-Rental-Listings | 17890 | 5 | 9 |
| 1415_beijing-pm2.5 | 43824 | 1 | 11 |
| 1649_Tamilnadu-Crop-production | 13266 | 4 | 3 |
| stats | 10000 | 1 | 9 |
| 1466_post-operative | 65532 | 1 | 11 |
| Airline_Delay_Cause | 100000 | 4 | 17 |
| 1704_House-Rent-in-Indian-Cities-and-Lo calities | 10692 | 5 | 8 |
| credit_card_defaulter | 10000 | 2 | 2 |
| 1781_SDSS-16 | 100000 | 1 | 17 |
| 2131_houses | 20640 | 0 | 9 |
| 1595_Oranges-vs.-Grapefruit | 10000 | 1 | 5 |
| 1140_exercises | 15000 | 1 | 6 |
| 1245_Production-cross-sections-of-Inert -Doublet-Model | 50625 | 0 | 13 |
| 2136_nyc-taxi-green-dec-2016 | 100000 | 0 | 10 |
| 0684_BNG(autoPrice) | 100000 | 0 | 16 |
| 1904_Apple-Complete-Stock-Data1980-2020 | 10015 | 0 | 6 |
| 1107_rainfall_bangladesh | 16755 | 2 | 2 |
| 2659_video_transcoding | 68784 | 2 | 17 |

Table 7: Regression Task Files

# E NUMERICAL RESULTS FOR UTILITY

| Model | 30 | 50 | 100 | 200 | 500 | Full |
|---|---|---|---|---|---|---|
| Real | 0.79 | 0.79 | 0.82 | 0.83 | 0.88 | 0.90 |
| SMOTE | 0.67 | 0.74 | 0.76 | 0.77 | 0.83 | 0.85 |
| PATECTGAN | 0.31 | 0.27 | 0.32 | 0.34 | 0.37 | 0.40 |
| AIM | 0.44 | 0.48 | 0.55 | 0.62 | 0.52 | 0.57 |
| CTGAN | 0.41 | 0.52 | 0.52 | 0.54 | 0.63 | 0.64 |
| GReaT | - | - | 0.76 | 0.80 | 0.84 | 0.85 |
| TVAE | 0.75 | 0.77 | 0.79 | 0.79 | 0.82 | 0.84 |
| TabDDPM | 0.77 | 0.79 | 0.81 | 0.81 | 0.84 | 0.85 |
| TabSyn | 0.76 | 0.78 | 0.81 | 0.82 | 0.85 | 0.86 |
| CTSyn | 0.79 | 0.81 | 0.83 | 0.84 | 0.84 | 0.86 |

Table 8: ML utility for classification benchmarks.Columns represent training examples(shots) provided.

| Model | 30 | 50 | 100 | 200 | 500 | Full |
|---|---|---|---|---|---|---|
| Real | 0.24 | 0.23 | 0.21 | 0.17 | 0.15 | 0.14 |
| SMOTE | 0.48 | 0.24 | 0.22 | 0.16 | 0.13 | 0.11 |
| AIM | 10274.55 | ≫10k | ≫10k | ≫10k | ≫10k | 110.14 |
| PATECTGAN | ≫10k | ≫10k | ≫10k | ≫10k | ≫10k | ≫10k |
| CTGAN | 0.97 | 0.25 | 0.28 | 0.19 | 0.21 | 0.20 |
| TVAE | 0.60 | 0.50 | 0.47 | 0.33 | 0.34 | 0.28 |
| GReaT | 0.30 | 0.25 | 0.23 | 0.20 | 0.19 | 0.16 |
| TabDDPM | 0.35 | 0.30 | 0.27 | 0.21 | 0.20 | 0.18 |
| TabSyn | 0.27 | 0.23 | 0.22 | 0.19 | 0.16 | 0.13 |
| CTSyn | 0.22 | 0.18 | 0.17 | 0.15 | 0.14 | 0.12 |

Table 9: ML utility for regression benchmarks. Columns represent training examples(shots) provided.

# F VAE FOR TEXT AND DATETIME VARIABLES

In our main experiment, we omitted free text and date-time variables to align with prior research in tabular transfer learning (Yan et al., 2024). However, CTSyn can handle these data types with proper decoder. We conducted preliminary VAE reconstruction experiments on the 'ebay reviews' and '0875 nfl games' table from OpenTab, incorporating textual and date-time columns. We train a VAE model from scratch jointly on these two data frames. For text encoding and decoding, we followed (Lovelace et al., 2024), encoding text with a pre-trained frozen BART-base model (Lewis et al., 2020) followed by a 4 layer Perceive resampler encoder-decoder network, and train the entire VAE and text encoder with cross-entroly loss on token prediction. Date column is cyclically encoded (Guo & Berkhahn, 2016) to capture periodicity and decoded with a linear layer with circular loss function. The reconstruction performances are as follows:

| Metric | Numerical (MSE) | Categorical (Accuracy) | Text (ROUGE-1, eBay) | Date-Time (Circular Loss, NFL) |
|---|---|---|---|---|
| Score | 0.0005 | 92% | 0.80 | 0.021 |

Table 10: Performance metrics across different data types.

These findings demonstrate CTSyn's potential to handle diverse data types without compromising performance. Further studies can be explore the potential of text pre-training in assisting categorical reconstruction, as well as text decoding as a method for table summarization tasks.

## G    METADATA GENERATION

To generate metadata for each table, we used GPT-4o (gpt-4o-2024-08-06). The metadata serves as a textual description capturing the context of the table, aiding in cross-table learning and improving generalization across diverse tabular datasets. Each table's metadata was generated by providing GPT-4o with the first 10 rows of the dataset and prompting it to generate a concise yet informative description. Below, we detail the exact procedure, including an example prompt and the corresponding metadata generated for the *Indian Liver Patient Dataset*.

### G.1    QUERYING GPT-4O FOR METADATA

For each table, we queried GPT-4o using a structured prompt. The prompt included:

- A brief instruction specifying that the model should describe the table.
- The first 10 rows of the dataset, formatted as a tab-separated table.
- A request for a summary that captures the table's context.

### G.2    EXAMPLE PROMPT

The following prompt was used for query GPT-4o in metadata generation:

```
You are analyzing a dataset called '{dataset_name}'.
Below are the first 10 rows of this dataset along with the corresponding
column names:
  Columns: {column_names}
  Data: {first_row_of_data_frame_in}

Based on the above data, please do the following:
1. Identify which columns are likely categorical.
2. Determine whether all categorical columns are represented in text form.
   If a categorical column has integer values like 0 or 1, then it is not
   in text form. But if a categorical column has values like male/female,
   then it is in text form.
3. Output the result as a JSON string with:
   - "dtypes": A dictionary mapping column names to "categorical" or
     "numerical".
   - "metadata": A short description of the dataset background.
   - "nontextcolumns": A list of categorical columns not in text form.

The output should be a JSON string and should not contain anything else.
```

### G.3    GENERATED METADATA EXAMPLE

Upon submitting this prompt to GPT-4o, the model generated the following metadata:

> "The dataset contains numerical data related to liver function and alcohol consumption. It is likely used for medical research purposes, particularly to analyze the effects of alcohol consumption on liver-related enzyme levels and overall liver health. Each column represents specific biochemical parameters from liver tests as well as the reported number of alcoholic drinks consumed by an individual."

## H    LATENT SPACE VISUALIZATION

We conducted a qualitative analysis comparing the encodings generated by the Transformer-VAE used in TabSyn with those from the encoder module of our VAE on an insurance dataset. Latent vectors of data points were color-coded based on categories. The results demonstrate that our model maintains clear separation of groups in the encoding space, providing evidence that our shared latent space enhances model capacity while maintaining interpretability for heterogeneous tables.

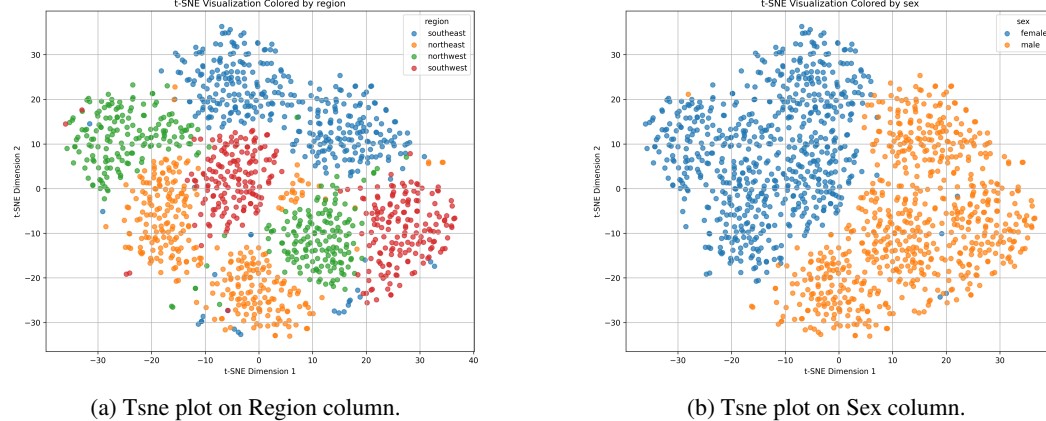

(a) Tsne plot on Region column.  (b) Tsne plot on Sex column.

Figure 4: Tsne visualization of embedding space created by pre-trained VAE model, on columns of Insurance dataset.

## I  ENCODING UNINFORMATIVE FEATURES

| Metric | Original Data | Uninformative Column Names |
|---|---|---|
| Shape | 0.94 (0.02) | 0.92 (0.03) |
| Corr | 0.95 (0.02) | 0.93 (0.03) |
| Precision | 0.64 (0.04) | 0.60 (0.05) |
| Recall | 0.075 (0.006) | 0.070 (0.007) |

Table 11: Comparison of metrics between Original Data and Uninformative Column Names.

Non-informative or abbreviated column headers(such as "X1", "VAR1") is a notable concern for tabular foundation models, especially as such models are expected to be pre-trained on web-scale dataset with limited curation in a self-supervised manner. To evaluate the robustness of CTSyn against such noisy data, we conducted an experiment where all column names in downstream benchmarks used during finetuning were replaced with single English letters (A–Z). This setting simulates real-world scenarios with non-descriptive headers in finetuning data. The results show a slight decline in statistical fidelity; however, CTSyn maintains a high level of performance, demonstrating its robustness to uninformative column headers.

## J  COMPUTATION

Our training are completed on an Amazon AWS g5.12xlarge instance, with 192 GB system memory, 4 Nvidia A10G GPU with $4 \times 24$ GB GPU memory. The pre-training time of CTSyn, GReaT and TabSyn are shown in the table 12.

| Model | VAE | Generation |
|---|---|---|
| CTSyn | 12 hours | 12 hours |
| GReaT | - | 50 hours |
| TabSyn | $86 \times 0.5 = 43$ hours | 24 hours |

Table 12: Pre-training computation cost. Note that TabSyn requires training table-specific autoencoders.

