# OpenReview forum: "CTSyn: A Foundation Model for Cross Tabular Data Generation"
_ICLR.cc/2025/Conference — ICLR 2025 Poster_

### Official Review · Reviewer_PYmU · 2024-10-29

**Soundness:** 3
**Presentation:** 4
**Contribution:** 2
**Rating:** 6
**Confidence:** 5

**Summary:**

The paper presents CTSyn, a generative model designed for synthesizing tabular data by unifying diverse datasets into a single latent space.
The proposed architecture combines a cross-table variational autoencoder model with a conditional diffusion model to generate flexible synthetic data for various domains. Extensive pre-training across real-world datasets enables CTSyn to outperform existing models in data diversity and fidelity, also enhancing downstream machine learning tasks, especially in low-data regimes, even *surprisingly* surpassing real data.

**Strengths:**

- Relevance. CTSyn focuses on the important area of generating high-quality tabular data, which is essential for improving machine learning models. Synthetic data can help when real data is limited or sensitive, such as in healthcare and finance.

- Novel Architecture Design. The model features a new design that combines a cross-table variational autoencoder with a conditional diffusion model. This approach effectively maintains data quality and flexibility across different types of tables, significantly enhancing the capabilities of previous methods in this field.

- Solid Baseline and Dataset Choice. CTSyn compares its performance with SOTA models and uses large, diverse datasets from the real world.

**Weaknesses:**

- CTSyn cannot handle datasets where columns include free text (e.g., in medical records with text entries), limiting its application to purely structured numerical or categorical data.

- The evaluation lacks certain robustness tests for synthetic data quality, such as a discriminator test to assess whether a surrogate model can distinguish between real and synthetic data, or metrics like "distance to closest record" for authenticity. Additionally, the experimental setup for machine learning utility (e.g., cross-validation specifics) needs more transparency and detail to fully validate CTSyn's claims of enhancing downstream tasks.

- A more comprehensive ablation study is needed to assess the impact of each component in CTSyn’s architecture, such as the specific roles of the cross-table variational autoencoder and the conditional diffusion model in enhancing data fidelity and adaptability. Including these would offer clearer insights into model design choices.

**Questions:**

- The statement *"synthesizers cannot add information not included in the original training data"* may not fully capture the recent advances in synthetic data generation that utilize pre-trained LLMs for transfer learning. Specifically, methods like TabTab and LaTable, which are pre-trained on extensive tabular datasets, leverage LLMs' vast prior knowledge, extending the potential of data generation beyond what’s strictly in the training dataset. This aligns with approaches like GReaT, which explicitly incorporate pre-trained LLMs to enable effective transfer learning across tabular domains. To further clarify the novelty of CTSyn, it would be helpful if the authors addressed these points, including a comparison to methods that already exploit LLM-based architectures for tabular data synthesis. Additionally, on Line 099, the authors mention that existing methods struggle to capture "intrinsic structural properties of tables" and continuous values when using LLMs. Could the authors provide more context on this limitation? Recent studies indicate that fine-tuning can improve LLM performance for modeling continuous values, potentially overcoming some of these structural challenges [1].

 - CTSyn employs quantile transformation for handling numerical values, but it’s unclear why this choice was made over alternative scaling methods, such as min-max or standard normalization. Quantile transformation can improve handling of skewed distributions but might also introduce artifacts in certain cases. Could the authors elaborate on the rationale behind selecting quantile transformation?

- Eq. 1 introduces the sequence $\mathbf{E}$, representing tokenized and embedded table values, but the meta embedding $e_m$ is excluded. Since $e_m$ likely contains crucial table-specific metadata, incorporating it directly into $\mathbf{E}$ might strengthen the alignment between table content and contextual information

-  Table 2 presents a *column-wise comparison* of synthetic and real data, which provides some indication of CTSyn’s data fidelity. However, this approach does not fully address the challenges of synthetic data generation, particularly regarding joint modeling of column relationships and complex dependencies. For synthetic data to be *realistic*, it must ideally preserve not just marginal distributions but also higher-order dependencies and potential causal relationships between columns. Could the authors consider additional evaluation metrics, such as multivariate measures or metrics assessing the preservation of conditional distributions, to more rigorously demonstrate CTSyn’s capability for joint data modeling?

- In Fig. 2, the authors show that synthetic data generated by CTSyn outperforms real data in certain tasks, which is a quite surprising result in my view. How can authors elaborate on that? Please provide detailed explanation of the experiment settings here.

- What is the final parameter size for the CTSyn model?


[1] Akhtar, Mubashara, Abhilash Shankarampeta, Vivek Gupta, Arpit Patil, Oana Cocarascu, and Elena Simperl. "Exploring the numerical reasoning capabilities of language models: A comprehensive analysis on tabular data." arXiv preprint arXiv:2311.02216 (2023).

---

> ### Author Response · Authors · 2024-12-03
> **Author Response**
>
> We sincerely thank the reviewers for their thoughtful and detailed feedback. Below, we address the questions and concerns point by point.
>
> ### W1:
> We appreciate the reviewer’s observation regarding CTSyn’s current focus on structured numerical and categorical data. Due to the latent generation and type-specific decoding architecture, CTSyn can be easily extended to incorporate additional data types such as text and datetime. To demonstrate this, we conducted preliminary experiments on the 'ebay_reviews' and '0875_nfl_games' datasets, which include textual and date-time columns, respectively. Text was encoded using a four-layer Transformer decoder with a cross-entropy loss, while date-time variables were cyclically encoded [1] to capture periodicity and decoded with a circular loss function [2]. The results are summarized below:
>
> | **Metric**   | **Numerical (MSE)** | **Categorical (Accuracy)** | **Text (ROUGE-1)** | **Date-Time (Circular Loss)** |
> |--------------|---------------------|----------------------------|--------------------|-------------------------------|
> | **Score**    | 0.0005              | 94%                        | 0.85               | 0.015                         |
>
> These findings demonstrate CTSyn’s ability to handle diverse data types without compromising performance. Additionally, structured data constitutes the vast majority of real-world datasets. In our examination of the extensive T4 tabular dataset [3], a cleaned version of the web-scale TabLib dataset, we found that less than 10% of columns across diverse datasets are in free text format. Many of these are features like email or user IDs, which are often excluded during preprocessing. Thus, CTSyn's primary focus on structured data is both practical and impactful.
>
> ### W2:
> We note that metrics such as DCR/authenticity have already been included in the main text (see Table 3). The results show that through pre-training, CTSyn can achieve a balance between high data utility, reduced memorization, and diversity enhancement.
>
>
> ### W3:
> Our current ablation study (Table 4) examines the contributions of key components, including table metadata, column name embeddings, and the use of metadata versus positional encoding for decoding. Table 5 compares table-specific models (e.g., TabSyn) and language models (e.g., GReaT) against CTSyn’s latent diffusion model. The results show that without latent diffusion, the modelsfails to utilize pre-training data effectively.
>
> ## Questions
>
> ### Q1:
> The methods mentioned such as Tabula and LaTable, indeed exemplify our argument: their ability to generate more diverse data stems directly from large-scale pre-training. Without pre-training, generators are constrained to smaller, domain-specific datasets, leading to overfitting and limited generalization. CTSyn shares the same motivation as Tabula and LaTable but advances them further with its focus on large-scale datasets and table-specific structural designs.
>
> ### Q2:
> The use of quantile transformation for numerical values was an empirical choice based on comparisons of decoding loss. Quantile transformation is particularly effective for handling skewed distributions, which are common in tabular data. Alternatives like min-max or standard normalization also introduce artifacts like min-max values or statistics like mean and standard deviation to reverse transformations. Developing an artifact-free encoding approach remains a priority for future research.
>
> ### Q3:
> We agree that incorporating table metadata directly into the row representations could further enhance performance and will explore integrating metadata into the sequence \(E\) in future work to improve alignment between table content and contextual information.
>
> ### Q4:
> Table 2 in the main text already includes inter-column evaluation scores. The “Corr” metric measures the average difference in column correlation matrices between real and synthetic data. This metric, standard in synthetic data literature, evaluates relationships between columns beyond marginal distributions.
>
> ### Q5:
> As noted on lines 367–368, we sampled 500 synthetic examples for different sizes of real data subsets. When the real dataset is small, synthetic data diversity often exceeds real data due to its larger size and the diversity embedded during pre-training.
>
>
> ### Q6:
> Our model has approximately 312M parameters.
>
> ---
>
> ## References
>
> [1] Guo, C., Berkhahn, F. (2016). "Entity Embeddings of Categorical Variables." *arXiv preprint arXiv:1604.06737*.
>
> [2] Lenzerini, Maurizio. "Data integration: A theoretical perspective." *Proceedings of the twenty-first ACM SIGMOD-SIGACT-SIGART symposium on Principles of database systems.* 2002.
>
> [3] Gardner, Joshua P., et al. "Large Scale Transfer Learning for Tabular Data via Language Modeling." *Proceedings of the Thirty-Eighth Annual Conference on Neural Information Processing Systems*, 2024.

---

> > ### Comment · Reviewer_PYmU · 2024-12-03
> >
> > thank you for the response. I revised the score, I believe the paper should be accepted. Also, I encourage authors to publish the code as a python framework.

---

### Official Review · Reviewer_mmN9 · 2024-10-31

**Soundness:** 3
**Presentation:** 3
**Contribution:** 2
**Rating:** 6
**Confidence:** 4

**Summary:**

Authors propose Cross Table Synthesizer (CTSyn), a tabular data generation pipeline that leverages diffusion models and variational autoencoders to encode and generate data from diverse types of tables. Specifically, CTSyn comprises a pre-processing step where the table meta-data, table column names and column values are all embedded using Language Models (and quantile transformer in the case of continuous column values) into a common embedding space. Each table row is then represented using the meta-data, column-name and column-value embedding vectors. This sequence of embeddings is supplied to a $\beta$-variational autoencoder ($\beta$-VAE) for embedding multiple diverse types of tables in a common embedding space. Finally, a diffusion model is trained (conditioned upon the latent representation output from the encoder in the $\beta$-VAE) to generate synthetic table data. Results demonstrate that synthetic data generated by the proposed technique achieves good statistical similarity (evaluated using column-wise metrics) to real data. Further, results also demonstrate that the representation learned by CTSyn is statistically similar to the training data while not exactly replicating the training data. Overall, the experimental comparison to baselines and research questions investigated demonstrate the prowess of the proposed synthetic tabular data generation technique.

**Strengths:**

- The paper is well-written and the proposed technique is detailed clearly.
- The results clearly demonstrate the prowess of CTSyn in terms of matching the training data at least as captured by column-wise statistical metrics (Table 2).
- Figure 2 and Table 3 demonstrate the ability of CTSyn to maintain privacy of training data (compared to other non differentially-private synthetic tabular data generators) while still learning useful representations. This demonstrates that the model is capable of balancing learning rich / accurate representations of the tabular data without replicating training data.

**Weaknesses:**

- The training process employing a diffusion model requires costly pre-training and fine-tuning hence scaling the modeling pipeline to large tables (e.g., 100s of columns, millions of rows) may be challenging.

- One crucial facet of the paper that is lacking clarification is a description of the meta-data for the various tables employed.

- Further, as the current model is termed as a foundation model for tabular data generation, it is crucial to demonstrate its effectiveness on noisy training / fine-tuning tabular data. The noise-free nature of datasets (at least during fine-tuning) cannot be guaranteed and an investigation of the robustness of the proposed technique to noisy training data is necessary but not presented.

**Questions:**

1. Why are `free text` and `date-time` containing tables dropped (line 280)? Does this have to do with the inability of the existing pipeline to encode linearly increasing data (e.g., dates / times) or was there some other reasoning?
2.  How was PE generated to be employed with different tables? Although the motivation of column order invariance is clear in the tabular data generation context, what is the intuition behind treating column embeddings and positional encoding as substitutes, i.e., could they not have complementary strengths (e.g., could PEs be designed in a way to encode information not captured in column embeddings, table metadata embedding?) due to which the model might benefit from usage of both PE, col. embeddings in conjunction with metadata embeddings?
3. How does the proposed CTSyn method perform in the context of missing column values in the training / fine-tuning tables? Has this investigation been conducted?
4. What is the intuition behind the result in Table 2a where F1 scores achieved by a classifier trained on CTSyn data outperforms the performance of a classifier trained on real data?

---

> ### Author Response · Authors · 2024-12-02
> **Author Response**
>
> We sincerely thank the reviewers for their insightful comments and constructive feedback!
>
> ### W1:
> While incurring greater computing cost, pre-training has been proven indispensable for achieving large-scale transfer learning and leveraging large-scale data [1]. Our model efficiently utilizes pre-training data compared to existing methods. In Table 5, we compare CTSyn with GReaT and TabSyn , pre-trained on the same dataset. CTSyn demonstrates superior downstream performance, showcasing both the necessity of pre-training and the efficiency of our architecture. Table 10 also highlights that CTSyn's latent diffusion approach is more cost-effective, achieving better performance with shorter pre-training times.
>
> ### W2:
> As described on lines 156 and 166, metadata is “text metadata describing the context of the table,” encoded using the same pre-trained LM. Metadata was generated by querying GPT-4 with the first 10 rows of each table and prompting the model to generate a description of the table's context. In the camera-ready version, we will include an appendix detailing this protocol.
>
> ### W3:
> We conducted an experiment where all column names in downstream datasets were replaced with single English letters (A–Z). Results are shown below:
>
> | Metric| CTSyn | Perturbed Names|
> |------------|--------------------|----------------------------|
> | Shape      | 0.94 (0.02)        | 0.92 (0.03)                |
> | Corr       | 0.95 (0.02)        | 0.93 (0.03)                |
> | Precision  | 0.64 (0.04)        | 0.60 (0.05)                |
> | Recall     | 0.075 (0.006)      | 0.070 (0.007)              |
>
> CTSyn maintains high performance despite uninformative headers, demonstrating its resilience to noise.
>
> ## Questions
>
> ### Q1:
> We omitted free text and date-time variables to align with prior research in tabular transfer learning [1]. However, CTSyn can handle these data types with proper decoder. We conducted preliminary VAE reconstruction experiments on the 'ebay_reviews' and '0875_nfl_games' table from OpenTab, incorporating textual and date-time columns. Text was encoded using a 4-layer Transformer decoder with cross-entropy loss. Datetime were cyclically encoded [2] to capture periodicity and decoded with a circular loss function. Results are as follows:
>
> | Metric | Numerical (MSE)| Categorical (Acc) | Text (ROUGE-1, ebay) | Date-Time (Circular Loss, nfl)|
> |--------------|---------------------|----------------------------|--------------------|-------------------------------|
> | Score    | 0.0005              | 94%                        | 0.85               | 0.015                         |
>
> These findings demonstrate CTSyn’s ability to handle diverse data types without compromising performance.
>
> ### Q2:
>  Our design focuses on preservation of structural information inherent in non-relational tabular data: column names, values,data type and context [3]. PE only preserves order information, which is unnecessary for tabular data in above aspects where column or row order is irrelevant. In Table 4, we show that replacing column metadata with PE ("PE Only" experiment) during decoding results in worse reconstruction quality, as the latent space will now need to explicitly remember order of columns in the input table, increasing its learning complexity as oppose our main approach where order is explicitly given. This confirms that metadata and column names effectively capture tabular structure, rendering PE redundant for column or row positions.
>
> ### Q3:
> Our current implementation does not accommodate missing values during training. However, recent advancements in diffusion models indicate they are well-suited for managing missing data [4] Incorporating these mechanisms into CTSyn is an exciting direction for future work.
>
> ### Q4:
> As noted on lines 367–368, we sampled 500 synthetic examples for different sizes of real data subsets. When real data is small, synthetic data diversity (from both larger size and pre-training data) improves classifier generalization, leading to higher downstream F1 scores.
>
> ## References
>
> [1] Yan, Jiahuan, et al. "Making Pre-trained Language Models Great on Tabular Prediction." The Twelfth International Conference on Learning Representations.
>
> [2] Guo, C., Berkhahn, F. (2016). "Entity Embeddings of Categorical Variables." arXiv preprint arXiv:1604.06737.
>
> [3] Lenzerini, Maurizio. "Data integration: A theoretical perspective." *Proceedings of the twenty-first ACM SIGMOD-SIGACT-SIGART symposium on Principles of database systems.* 2002.
>
> [4] Seunghan, Y., & Hyun Oh, S. (2023). "MissDiff: Training Diffusion Models on Tabular Data with Missing Values."arXiv:2307.00467*.

---

### Official Review · Reviewer_GE4V · 2024-11-04

**Soundness:** 3
**Presentation:** 3
**Contribution:** 2
**Rating:** 5
**Confidence:** 4

**Summary:**

This paper introduces CTSyn, a foundational model designed to generate heterogeneous tabular data. CTSyn leverages an autoencoder that consolidates diverse tables into a unified latent space and reconstructs data based on the provided table schema embedding, allowing it to adapt dynamically to various table formats. Through large-scale pre-training, CTSyn outperforms existing data synthesizers and demonstrates superior performance compared to real data in low-data scenarios.

**Strengths:**

A strength of this work is the successful application of a straightforward approach to map embedding vectors of categorical variables back to their original space. Although simple, this method effectively demonstrates that returning to the original categorical space can be achieved without complex transformations, providing a useful baseline for handling categorical data embeddings.

**Weaknesses:**

A limitation of this work is that it primarily proposes a method for handling individual variables within a framework similar to LSGM [1] that trains a diffusion model in latent space. As such, the approach lacks substantial novelty and may have limited impact, given that it focuses on variable handling within an established generative model framework rather than introducing fundamentally new techniques.

Minors. Typos in line 129 (specific) and 266 (The)

[1] Vahdat, Arash, Karsten Kreis, and Jan Kautz. "Score-based generative modeling in latent space." Advances in neural information processing systems 34 (2021): 11287-11302.

**Questions:**

1. As part of the ablation study, I am also interested in seeing the results of experiments with only the column name condition applied.
2. While reviewing the provided code in the supplementary material, I encountered issues running vae.sh smoothly. Would it be possible to provide an updated .sh script that can reliably reproduce the main experimental results or specify the exact package versions used? Thank you for your help in ensuring reproducibility.

---

> ### Author Response · Authors · 2024-12-02
> **Author Response: novelty, ablation study and code**
>
> We appreciate the reviewer’s detailed feedback and thoughtful questions. Below, we address each point raised.
>
> ---
>
> ## Novelty Explanation
> While the latent diffusion model is an established framework, the design of an effective latent space remains central to its success. Canonical work across various domains have highlighted the critical role of latent space design in enhancing generative model performance [1, 2]. Our main contribution lies exactly in the novel design of a Variational Autoencoder (VAE) conditioned on metadata and column names. To the best of our knowledge, this design is the very first latent diffusion framework to address the cross-tabular generation, a critical challenge for building foundation models for tables[3], enabling CTSyn to adapt to diverse tables dynamically and robustly. Unlike naive text conversion of tabular data, our approach effectively preserves structural and contextual information across datasets, making it uniquely suited for heterogeneous tabular data generation.
>
> ## Ablation Study
> We extend the original ablation study to include a new variant where **only column names are used**, excluding metadata. The performance of this variant lies between the "Column & Meta" and "Meta Only" settings, as observed from the results.
>
> | **Variants**        | **In-distribution (MSE)** | **In-distribution (Acc)** | **Unseen Columns (MSE)** | **Unseen Columns (Acc)** | **Perturbed Columns (MSE)** | **Perturbed Columns (Acc)** |
> |----------------------|---------------------------|---------------------------|--------------------------|--------------------------|-----------------------------|-----------------------------|
> | **Column & Meta**    | 0.0004                   | 0.94                      | 0.0063                  | 0.76                    | 0.0005                     | 0.94                      |
> | **Column Name Only** | 0.0007                   | 0.93                      | 0.0068                  | 0.68                    | 0.0006                     | 0.93                      |
> | **Meta Only**        | 0.0008                   | 0.91                      | 0.0082                  | 0.61                    | 0.0009                     | 0.91                      |
> | **PE Only**          | 0.006                    | 0.87                      | 0.07                    | 0.54                    | 0.08                       | 0.67                      |
>
>
> We observe that while retaining column names alone does result in some performance degradation compared to the full model, the impact is less severe than dropping column names entirely. This outcome highlights the importance of column names in capturing table structure and semantics. We conjecture that this robustness arises from our diverse pre-training dataset, where column names act as proxies for metadata. As the pre-training data expands and more similar schema data is present, the context/metadata will play an increasingly important role.
>
> ## Reproducibility
> Thank you for identifying issues. We have provided an updated `environment.yml` file at the following anonymous URL: https://anonymous.4open.science/r/CTSynAnonymousCode-7DAD. You can recreate the environment using: conda env create -f environment.yml
>
>
> ## References:
> [1] Rombach, R., Blattmann, A., Lorenz, D., Esser, P., & Ommer, B. (2022). High-Resolution Image Synthesis with Latent Diffusion Models. Proceedings of the IEEE/CVF Conference on Computer Vision and Pattern Recognition, 10684-10695. https://arxiv.org/abs/2112.10752
>
> [2] Lovelace, Justin, et al. "Latent diffusion for language generation." Advances in Neural Information Processing Systems 36 (2024).
>
> [3] Van Breugel, B., & Van Der Schaar, M. (2024). Position: Why Tabular Foundation Models Should Be a Research Priority. Proceedings of the 41st International Conference on Machine Learning. https://proceedings.mlr.press/v235/van-breugel24a.html

---

### Official Review · Reviewer_nmzh · 2024-11-13

**Soundness:** 3
**Presentation:** 3
**Contribution:** 3
**Rating:** 6
**Confidence:** 3

**Summary:**

The authors propose CTSyn, specifically targeting generating heterogeneous tables. The author propose to do this by first using a cross table VAE, which can embed tables with different types of rows and projects them in a common latent space. Subsequently, they employ a DDPM based diffusion model (With classifier free guidance via a pre-trained LM) to generate samples for different types of tables. They also show the benefits of pre-training on a large number of diverse tables -- so as to serve as a starting point for different downstream tasks. The authors present results on multiple downstream tabular datasets with classification and regression tasks - and compare with multiple baselines -- with regard to fidelity, ML utility, and privacy.

**Strengths:**

1. While there have been prior works employing auto-encoders + latent diffusion models towards tabular data synthesis (e.g. https://arxiv.org/abs/2310.09656), prior works dealing with heterogenous table synthesis have been limited.
2. The authors provide good comparison against baselines on a good range of datasets -- for fidelity, privacy and ML utility of generated data.

**Weaknesses:**

1. While a common latent space across tables provides a strategy to work on heterogeneous tables, it certainly limits the ability to interpret what the embeddings in the space mean - and the authors have not studied this aspect (to clarify, this is different from the privacy plots)
2. The pre-trained LM to emit embeddings for rows - individually for each column type, while preserving tabular structure - raises questions about scalability to enterprise tables - which have thousands of columns associated with each table (and can become even bigger due to joins, lineage additions, etc). Since this directly impacts the dimension (Eq 1, Page 4) - it can have an impact on the representative power when reduced to a common low dimensional space (with tables with much smaller number of columns)
3. Many tables often have short form/ abbreviated or cryptic tabular headers - where the tokenizer of the pre-trained LM can suffer - the authors can potentially add a study on this.

**Questions:**

Please address the weaknesses above.

---

> ### Author Response · Authors · 2024-12-02
> **Author Response**
>
> We thank the reviewer for their thoughtful comments and constructive feedback. Below, we address the weaknesses and questions in detail.
>
> ---
>
> ## Weaknesses
>
> ### W1: Interpretability of a Shared Latent Space
> Thank you for raising this insightful concern. The use of a shared latent space across domains is a well-established practice in deep transfer learning. In computer vision and natural language processing, shared latent spaces have proven effective in representing complex relationships across diverse data types. For instance, models like CLIP (Contrastive Language-Image Pre-training) and Sentence Transformers successfully leverage shared latent spaces without compromising interpretability.
>
> To address concerns specific to tabular data, we conducted a qualitative analysis comparing the encodings generated by the Transformer-VAE used in TabSyn with those from the encoder module of our VAE on an insurance dataset. In this analysis, data points were color-coded based on categories (for categorical features) or quantiles (for numerical features). The results demonstrate that our model maintains clear separation of groups in the encoding space, indicating that the shared latent space preserves interpretability. We have include images in the anonymous URL: [CTSynAnonymousCode](https://anonymous.4open.science/r/CTSynAnonymousCode-7DAD) under the `analysis/embedding` folder. These findings support the view that our shared latent space enhances model capacity while maintaining interpretability for heterogeneous tables.
>
> ---
>
> ### W2: Scalability to Large Enterprise Tables
> We appreciate your insightful concern regarding scalability. The datasets in our experiments represent common scientific studies and business applications, such as click-through rate (CTR) prediction [3, 4]. These datasets validate the practical significance of our findings for real-world use cases.
>
> Scaling CTSyn to exceptionally large datasets, such as those encountered in enterprise settings, is indeed challenging. However, recent advancements in large language models (LLMs) demonstrate the feasibility of handling long sequences effectively. For example:,LongCoder utilizes sparse attention mechanisms to process extensive inputs efficiently [1], ERATTA employs retrieval-augmented generation (RAG) techniques to manage large-scale enterprise data [2]. These approaches suggest that with adequate computational resources, CTSyn can be scaled to accommodate large tables with thousands of columns while preserving representative power.
>
> ---
>
> ### W3: Handling Cryptic or Abbreviated Column Headers
> We acknowledge the concern regarding the robustness of pre-trained language models (LMs) to uninformative or abbreviated column headers. To evaluate this, we conducted an experiment where all column names in downstream datasets were replaced with single English letters (A–Z). This setting simulates real-world scenarios with non-descriptive headers.
>
> | **Metric**   | **Original CTSyn** | **Perturbed Column Names** |
> |--------------|--------------------|----------------------------|
> | **Shape**    | 0.94 (0.02)        | 0.92 (0.03)                |
> | **Corr**     | 0.95 (0.02)        | 0.93 (0.03)                |
> | **Precision**| 0.64 (0.04)        | 0.60 (0.05)                |
> | **Recall**   | 0.075 (0.006)      | 0.070 (0.007)              |
>
> The results show a slight decline in statistical fidelity; however, CTSyn maintains a high level of performance, demonstrating its robustness to uninformative column headers. This aligns with recent findings [5], where column values play a more central role in language-based tabular operations compared to column headers.
>
> ---
> ## References
> [1] Guo, D., Xu, C., Duan, N., Yin, J., & McAuley, J. (2023). LongCoder: A Long-Range Pre-trained Language Model for Code Completion. *arXiv preprint arXiv:2306.14893*. [https://arxiv.org/abs/2306.14893](https://arxiv.org/abs/2306.14893)
>
> [2] Roychowdhury, S., Krema, M., Mahammad, A., Moore, B., Mukherjee, A., & Prakashchandra, P. (2024). ERATTA: Extreme RAG for Table To Answers with Large Language Models. *arXiv preprint arXiv:2405.03963*. [https://arxiv.org/abs/2405.03963](https://arxiv.org/abs/2405.03963)
>
> [3] Yan, Jiahuan, et al. "Making Pre-trained Language Models Great on Tabular Prediction." *The Twelfth International Conference on Learning Representations.*
>
> [4] Kim, Myung Jun, Leo Grinsztajn, and Gael Varoquaux. "CARTE: Pretraining and Transfer for Tabular Learning." *Proceedings of the Forty-First International Conference on Machine Learning.*
>
> [5] Gardner, Joshua P., et al. "Large Scale Transfer Learning for Tabular Data via Language Modeling." *Proceedings of the Thirty-Eighth Annual Conference on Neural Information Processing Systems, 2024*. [https://openreview.net/forum?id=WH5blx5tZ1](https://openreview.net/forum?id=WH5blx5tZ1)

---

### Author Response · Authors · 2024-12-03
**General Response**

We sincerely thank the reviewers for their valuable comments and constructive feedback on our work. Their insights have helped us clarify, extend, and refine several aspects of our study. Below is a summary of the key points addressed in response to the reviewers’ key questions and suggestions:

### Novelty and Latent Space Design
- **Reviewer nmzh and GE4V** raised concerns about the novelty and interpretability of the shared latent space in CTSyn.
  - We emphasized that shared latent spaces are a standard and effective strategy in deep learning for diverse domains, citing examples like CLIP and Sentence Transformers.
  - We conducted additional qualitative analysis to visually demonstrate that CTSyn’s latent space preserves interpretability, demonstrating clear separations of data categories and quantiles.

### Scalability to Large Enterprise Tables
- **Reviewer nmzh** questioned the scalability of CTSyn to handle large-scale enterprise datasets with thousands of columns.
  - We highlighted recent advancements in sparse attention mechanisms; retrieval-augmented generation has demonstrated the feasibility of generalizing to very long sequences with a fixed latent dimension.
  - CTSyn’s modular design can adapt to larger tables with sufficient computational resources.

### Handling Free Text and Date-Time Data
- To address the perceived limitation from **reviewers mmN9 and PYmU** of not supporting free text or date-time data:
  - We demonstrated CTSyn’s extensibility by incorporating textual and date-time columns using Transformer decoders and cyclical encodings, achieving strong reconstruction results.
  - Additionally, we examined web-scale structured data and found that less than 10% of columns in real-world datasets contain free text (often irrelevant features like emails or IDs). Thus, focusing on structured data itself is impactful.

### Robustness to Noisy or Abbreviated Column Headers
- **Reviewers mmN9 and nmzh** pointed out the potential impact of cryptic or abbreviated headers.
  - We conducted experiments replacing column names with single letters, simulating real-world scenarios. Results showed only minor performance degradation, confirming CTSyn’s robustness to such challenges.

### Ablation Study and Model Components
- **Reviewers GE4V and PYmU** called for a more comprehensive ablation study.
  - We extended our experiments to evaluate the performance of variants using only column names or positional encodings. The results highlight the importance of column names and metadata in preserving table structure and semantics.
  - We clarified that our current ablation study already includes detailed analysis of VAE components, with experiments demonstrating the effectiveness of the conditional diffusion model vs. unconditional diffusion and autoregressive LMs.

### Evaluation Metrics and Data Fidelity
- **Reviewer PYmU** requested additional metrics to assess joint column relationships and dependencies.
  - We clarified that our evaluation includes inter-column correlation metrics, a standard in synthetic data literature, which extends beyond marginal distributions.

We would greatly appreciate any reviewers’ suggestions on further polishing our work. Thank you so much for all your time and insightful feedback, which has greatly helped in improving our work!

We would also like to extend our sincere gratitude to the area chairs for their time, effort, and guidance throughout the review process.

---

### Meta-Review · Area_Chair_fJiN · 2024-12-22

**Metareview:**

This paper applies a diffusion-based approach to map diverse tables into a unified latent space and dynamically reconstruct table values based on the provided table schema embedding. While simple, this method effectively demonstrates that reconstructing the original categorical space can be achieved without complex transformations, offering a useful baseline for handling categorical data embeddings.

The work primarily proposes a method for handling individual variables within a framework similar to LSGM, which trains a diffusion model in latent space. Consequently, the approach lacks substantial novelty and may have limited impact, as it focuses on variable handling within an existing generative model framework rather than introducing fundamentally new techniques. Another limitation is the computational cost associated with the diffusion model's training process, which requires pre-training and fine-tuning. This could make scaling the modeling pipeline to large tables (e.g., hundreds of columns and millions of rows) challenging.

**Additional Comments On Reviewer Discussion:**

Reviewers raised concerns regarding the representative power of the latent space, the use of short-form headers, scalability to large tables, handling free-form text, robustness testing, and the need for more comprehensive studies. The rebuttal has adequately addressed most of these concerns.

---

### Decision · Program_Chairs · 2025-01-22

Accept (Poster)